# Major intensification of Atlantic overturning circulation at the onset of Paleogene greenhouse warmth

S.J. Batenburg [1,2], S. Voigt[1], O. Friedrich[3], A.H. Osborne[4], A. Bornemann [5], T. Klein[1], L. Pérez-Díaz[6] & M. Frank[4]

During the Late Cretaceous and early Cenozoic the Earth experienced prolonged climatic cooling most likely caused by decreasing volcanic activity and atmospheric $CO_2$ levels. However, the causes and mechanisms of subsequent major global warming culminating in the late Paleocene to Eocene greenhouse climate remain enigmatic. We present deep and intermediate water Nd-isotope records from the North and South Atlantic to decipher the control of the opening Atlantic Ocean on ocean circulation and its linkages to the evolution of global climate. The marked convergence of Nd-isotope signatures 59 million years ago indicates a major intensification of deep-water exchange between the North and South Atlantic, which coincided with the turning point of deep-water temperatures towards early Paleogene warming. We propose that this intensification of Atlantic overturning circulation in concert with increased atmospheric $CO_2$ from continental rifting marked a climatic tipping point contributing to a more efficient distribution of heat over the planet.

[1] Institut für Geowissenschaften, Goethe-Universität Frankfurt, Altenhöferallee 1, Frankfurt am Main 60438, Germany. [2] Department of Earth Sciences, University of Oxford, South Parks Road, Oxford OX1 3AN, UK. [3] Institut für Geowissenschaften, Ruprecht-Karls-Universität Heidelberg, Im Neuenheimer Feld 234-236, 69120 Heidelberg, Germany. [4] GEOMAR Helmholtz-Zentrum für Ozeanforschung Kiel, Wischhofstr. 1-3, Kiel 24148, Germany. [5] Bundesanstalt für Geowissenschaften und Rohstoffe, Stilleweg 2, 30655 Hannover, Germany. [6] Department of Earth Sciences, Royal Holloway, University of London, Egham TW20 0EX, UK. Correspondence and requests for materials should be addressed to S.J.B. (email: sietske.batenburg@earth.ox.ac.uk)

The Earth underwent long-term climatic cooling between the peak-greenhouse intervals of the mid-Cretaceous and the Eocene[1–5]. Globally averaged deep-water temperatures gradually declined by almost 10 °C from 72 to 59 Ma, as estimated from benthic foraminiferal oxygen-isotope data[3,6]. This cooling has been ascribed to decreasing atmospheric $CO_2$ levels[7–9] through global reduction of volcanism and sea-floor spreading rates[10] combined with changes in ocean circulation patterns[3]. In contrast, there is no comprehensive model explaining how the greenhouse conditions of the Eocene were established and what the roles of atmospheric $CO_2$ and ocean circulation were in promoting global warming. Mechanisms proposed so far have solely focussed on increased atmospheric $CO_2$ levels either induced by carbon cycle changes[6], rates of continental rifting[11], or by enhanced volcanism of the North Atlantic igneous province[12,13]. The role of changes in overturning circulation caused by the opening of the Atlantic Ocean and related changes in oceanic heat transport has, however, not been addressed yet.

While circum-equatorial flow, which had dominated circulation in the proto-North Atlantic earlier in the Cretaceous, gradually declined[14], the ongoing opening and deepening of the Atlantic basin[15,16] led to increased North-South connectivity, although the timing of the establishment of a deep-water connection remains debated[17–22]. Enhanced latitudinal water-mass exchange likely promoted the distribution of heat across the planet via the thermohaline conveyor and resulted in reduced temperature contrasts between the equator and the poles. To distinguish tectonic constraints on circulation from climatically driven changes, the role of subsiding submarine barriers has to be assessed. We determine the timing of the establishment of a persistent deep-water connection between the North and South Atlantic by combining deep-water neodymium (Nd) isotope and temperature records.

Assessing the role of ocean circulation on Earth's climate in the latest Cretaceous and early Paleogene requires tight constraints on the modes and locations of deep-water formation and the extent of mixing of different deep-water masses. Information on past water mass mixing and exchange can be derived from Nd-isotope signatures ($^{143}$Nd/$^{144}$Nd, expressed as $\varepsilon_{Nd(t)}$) of authigenic, seawater-derived sedimentary archives such as ferromanganese coatings of sediment particles or fish debris, which have been demonstrated to incorporate the Nd-isotope composition of ambient deep waters[23]. Deep-water masses mainly acquire their Nd-isotope signatures from continental contributions via rivers and dust inputs in their source areas[23], as well as through exchange processes with ocean margin sediments[24]. These characteristic Nd-isotope compositions of deep-water masses are then conservatively advected and mixed over large distances in the open ocean given that the average Nd residence time of 400–2000 years is similar to the global ocean mixing time[23,25]. Analysis of the Nd-isotope composition of authigenic sedimentary archives thus allows the reconstruction of changes in deep-water mixing over time.

Existing Late Cretaceous Atlantic seawater $\varepsilon_{Nd(t)}$ signatures display a large spread in values (Fig. 1) that led to the suggestion that different mechanisms and locations of deep-water formation operated simultaneously[21,26–30]. There are indications that intermediate and deep-water exchange commenced as early as at 90 Ma in the Late Turonian[20], although the deep Atlantic Ocean potentially operated as a number of sub-basins with limited connectivity until the Maastrichtian[22]. The large variability in Cretaceous $\varepsilon_{Nd(t)}$ values has so far been interpreted to reflect different modes and

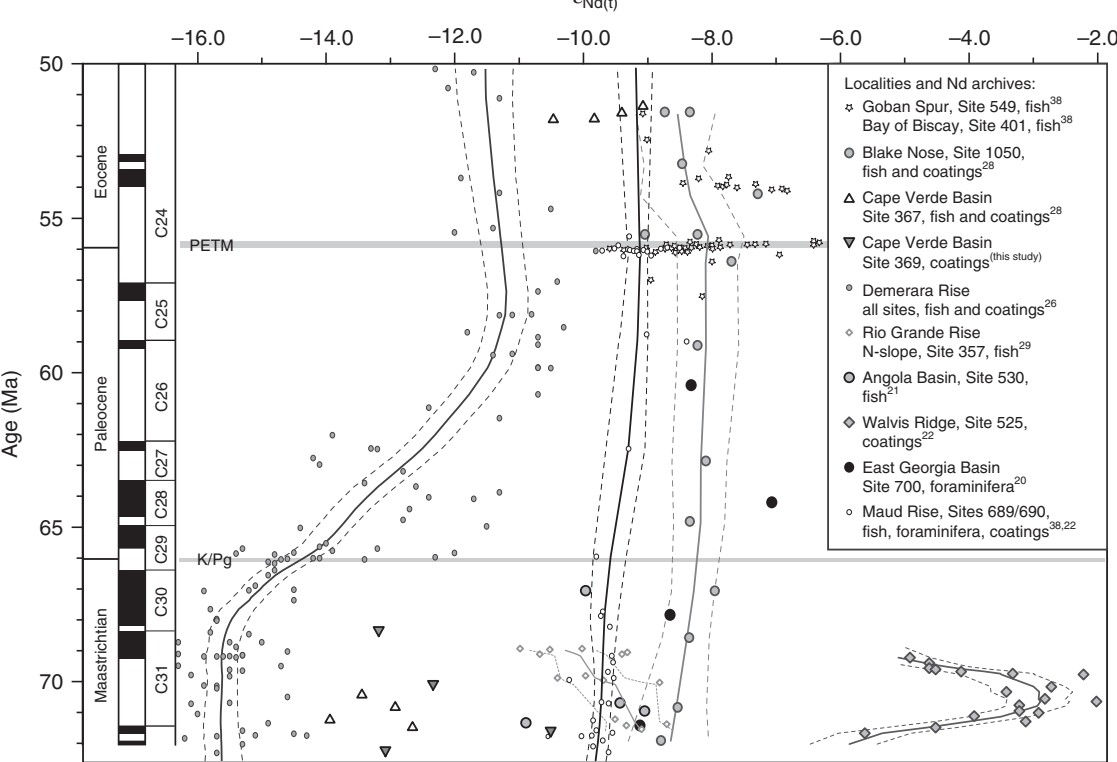

**Fig. 1** Nd-isotope data across the latest Cretaceous – early Paleogene. In this compilation of published Atlantic $\varepsilon_{Nd(t)}$ records and new Nd-isotope data from Site 369 (this study) only records with four or more data points over the time-interval 72–50 Ma are included. The $\varepsilon_{Nd(t)}$ data of Demerara Rise (Sites 1258, 1260, 1261), Maud Rise, Blake Nose and Site 525 have been smoothed by LOWESS regression ($f = 0.3$) with the dashed lines representing the 97.5% confidence interval. The $\varepsilon_{Nd(t)}$ data are flanked by the geomagnetic polarity time scale with chrons C31-C24. PETM: Paleocene/Eocene thermal maximum, K/Pg: Cretaceous/Paleogene Boundary

locations of deep-water formation either in the southern high latitudes[21,22,29,30] or in the North Atlantic[26–28], or local deep-water formation in relatively shallow sub-basins separated at depth[22]. The spread in $\varepsilon_{Nd(t)}$ values was likely further enhanced by local boundary exchange and weathering inputs into the relatively small Atlantic basins. Since the Cretaceous Atlantic Ocean was limited in depth and width, its contact area with the margins was large compared to its volume. Regional processes such as boundary exchange thus had a profound effect on water-column chemistry, as is the case for modern near-shore settings[24,31] or restricted sub-basins[32]. The Cretaceous Nd-isotope records from Demerara Rise and the Cape Verde Basin exemplify this effect by local weathering inputs of highly unradiogenic Nd from the old cratons of South America and Africa[27,28] (Fig. 1). Despite the potential influence of continental inputs near ocean margins, several open-ocean sites in the Late Cretaceous North and South Atlantic show parallel Nd-isotope trends. These parallel trends have been interpreted to reflect the formation and northward flow of a southern-sourced deep-water mass, "Southern Component Water"[30], although the behaviour of individual $\varepsilon_{Nd(t)}$ records is highly variable on a time scale of millions of years and patterns of change are dissimilar between localities.

From the Paleocene–Eocene Thermal Maximum (PETM) at 56 Ma onwards, most open-ocean $\varepsilon_{Nd(t)}$ signatures from the North and South Atlantic were within a narrow range of -8 to -10[28,33,34], indicating common water masses at bathyal and abyssal depths. There is, however, a lack of data for the Paleocene, which limits our understanding of when and to what extent deep waters exchanged and when the Atlantic started to play a key role in hemispheric oceanic heat exchange. A compilation of existing $\varepsilon_{Nd(t)}$ records for the period of time from 72 to 50 Ma (Fig. 1) shows that Nd-isotope data for the Paleocene are only available from a limited number of sites and, with the exception of Demerara Rise[26,28], are of limited resolution (less than one sample per two million years). The Paleocene, however, marks the time when the Atlantic significantly widened and deepened, which potentially paved the way for similar-to-modern ocean overturning processes[17]. Here we fill this gap and present new Paleocene intermediate- and deep-water Nd-isotope records from the North and South Atlantic Ocean. Five ocean drilling sites were selected from paleo-water depths between 500 and 4500 m (Supplementary Table 1/Fig. 2) to obtain seawater Nd-isotope records covering the critical time span from the end-Cretaceous to the early Paleogene.

## Results

### Seawater origin of Nd-isotope signatures.
Seawater Nd-isotope signatures were obtained by leaching ferromanganese coatings of bulk sediments that are considered a reliable archive if sufficiently weak leaching procedures are applied[35]. The $\varepsilon_{Nd(t)}$ variability of the detrital material was also determined for selected samples in this study (details in "methods" section), to evaluate the potential influence of local weathering inputs. The $\varepsilon_{Nd(t)}$ signatures of the detrital fractions and the leached ferromanganese oxide coatings show similar long-term trends at Sites 516 and U1403 and parts of the records at Sites 1267 and 525. Despite following parallel trends, most detrital $\varepsilon_{Nd(t)}$ values are significantly offset from those of the coatings supporting the validity of the seawater $\varepsilon_{Nd(t)}$ signatures extracted from the coatings at the offshore locations of our studied sites as faithful recorders of past water mass mixing (Fig. 3). The Nd-isotope composition of the water-masses themselves may have been influenced to some extent by local factors such as boundary exchange processes that mainly occur when deep-water circulation is slow and/or the sites were located in small or partly isolated basins with high detrital input[31,32,36].

In addition, the dissolved seawater Nd-isotope signature may have been incorporated into the hydrogenous component of pelagic clays[20,37], which may partly explain the similarity in the long-term evolution of the detrital and leached $\varepsilon_{Nd(t)}$ values.

### Parallel trends and convergence of Nd-isotope values.
Our new seawater Nd-isotope records from the North and South Atlantic (Fig. 3 and Supplementary Tables 2 to 6) display a wide range of values (−2 to −11) in the Maastrichtian interval (72.1–66 Ma) with parallel trends that converge to a common value of -8 to -9 at 59 Ma (Fig. 4). Our North Atlantic record from Site U1403 ends at 58 Ma, but $\varepsilon_{Nd(t)}$ values between −9.2 and 8 around 57 Ma at northern Site 549[38] corroborate our findings (Fig. 1).

Sites 525, 1267 and 516 in the South Atlantic, and Site U1403 in the North Atlantic show a trend of decreasing $\varepsilon_{Nd(t)}$ from approximately 70 to 63 Ma, with lowest values reached in the first half of the Paleocene. This decrease may reflect the reduction in active volcanism and exposed volcanic terrains in and around the Atlantic Ocean[20]. Nd-isotope values at Site 525 were positively offset from $\varepsilon_{Nd(t)}$ signatures at comparably shallow Site 516 on the Rio Grande Rise and nearby deeper Site 1267 at the base of the north-western slope of the Walvis Ridge until the end of the Cretaceous. This positive offset was most likely caused by the weathering influx of volcanic material from the partially subaerially exposed Walvis Ridge in the latest Cretaceous[15,39]. The offset decreased as the ridge and Site 525 subsided.

From approximately 64 Ma onwards, average $\varepsilon_{Nd(t)}$ values display an increasing trend until 60–59 Ma. We assign this trend to the enhanced volumetric flow of deep and intermediate water masses in the opening South Atlantic Basin which likely led to a decrease of the influence of local inputs and boundary change effects. In addition, the observed trend coincides with a first phase of magmatic activity of the North Atlantic Igneous Province from 62 to 61 Ma[13], which may have supplied radiogenic Nd, and ongoing deepening of the study sites that may have reduced unradiogenic weathering inputs from nearby continents.

From 59 Ma onwards, the Nd-isotope signatures at all newly studied sites, as well as Demerara Rise[26], decrease together and our $\varepsilon_{Nd(t)}$ results fall within a narrow range of -7 to -9.5 for the period 58.5–56.5 Ma. This convergence may reflect increasing admixture of southern-sourced deep water, which would have carried a $\varepsilon_{Nd(t)}$ signature similar to that at Maud Rise of approximately -9 in the Paleocene (Fig. 4)[20].

### Opening of the Atlantic Ocean and climatic implications.
Recent paleobathymetric reconstructions show that deep oceanic basins in the Atlantic Ocean, like the Cape and the Angola basins, were constricted until the end of the Cretaceous[15]. Deeper structures, such as the Vema and Hunter channels flanking Rio Grande Rise only allowed intermediate-water exchange at depths shallower than 2500 m. In the Paleocene, the South Atlantic deepened and widened, with the western portion of the Rio Grande Rise having subsided below 2500 m water depth at 60 Ma and the Argentine and Brazil basins reaching depths of over 5500 m in the early Eocene[15,40] (Fig. 2). The close correspondence in $\varepsilon_{Nd(t)}$ signatures at 59 Ma suggests a common deep-water signature ($\varepsilon_{Nd(t)}$ −9 to −8) in the South and North Atlantic (Fig. 4). We interpret the converging trend of Nd-isotope signatures to reflect an increasingly efficient deep-ocean circulation in the Atlantic Ocean with the dominant deep-water masses most likely originating in the high southern latitudes. Such a southern origin of deep water is consistent with recent modelling results[17] suggesting locations of major deep-water formation in the Southern Ocean, potentially supplemented by a minor source of deep water formed offshore North America. At the same time, our data show

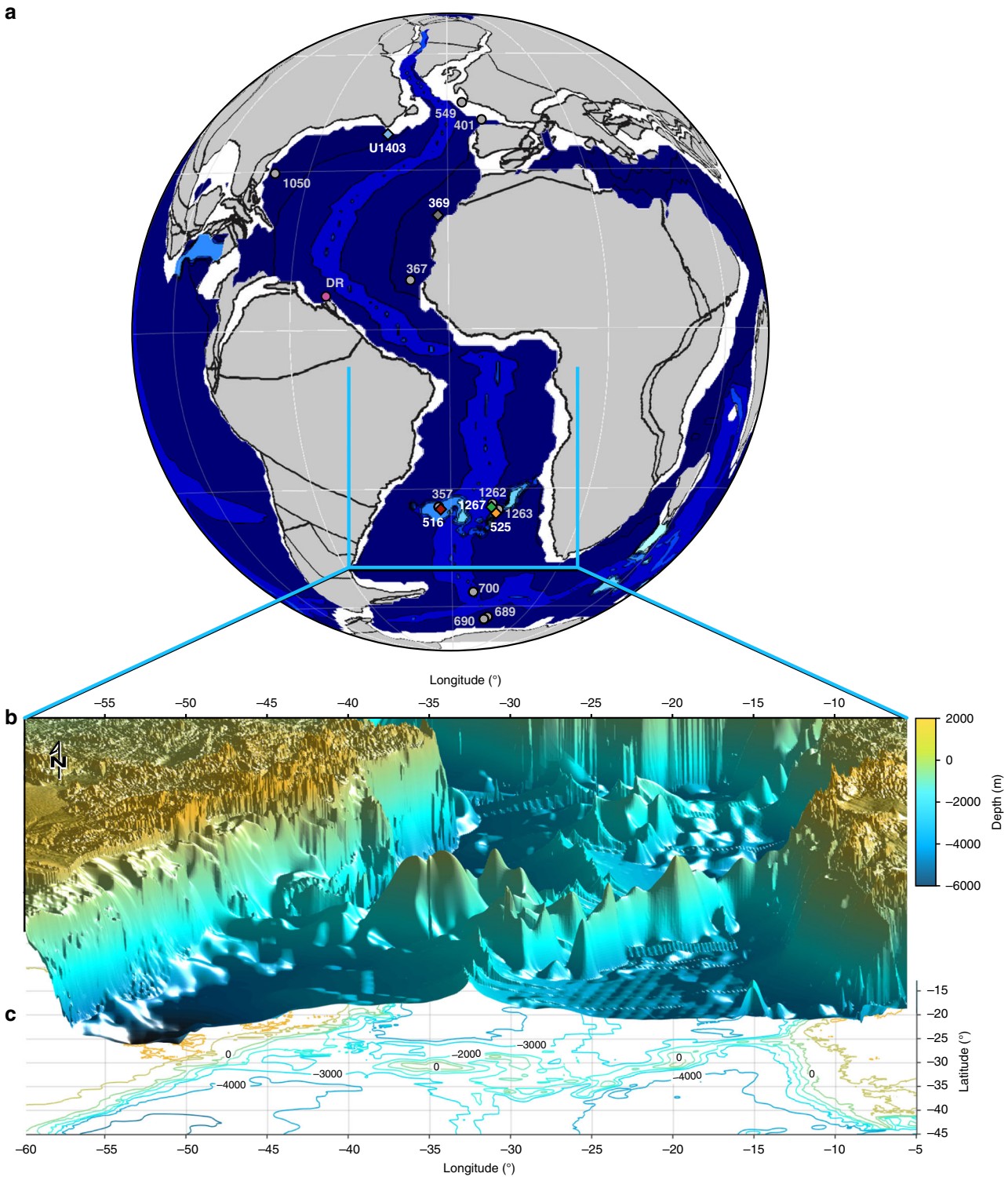

**Fig. 2** Paleogeographic setting. **a** Paleogeographic reconstruction at 60 Ma. Plate tectonic configuration after ref. [63] and global bathymetry from ref. [39]. Circles indicate Atlantic Ocean drilling sites for which Nd-isotope data are available (at least four points in the time-interval 72–50 Ma), DR: Demerara Rise. Diamonds indicate locations for which new data are presented here. Numbers indicate DSDP/ODP/IODP Sites. **b** Three-dimensional plot of the reconstructed bathymetry of the South Atlantic with the Rio Grande Rise – Walvis Ridge barrier at 60 Ma from ref. [40]. **c** Contour plot of the bathymetry in panel B, with black numbers indicating depths in meters

that the sub-basins of the deep Atlantic became fully connected by subsidence of the Rio Grande Rise near 59 Ma, accompanied by the widening and deepening of the equatorial gateway[17,20]. The improved connectivity and the increased volumetric exchange of water masses in the Atlantic Ocean at 59 Ma allowed

modern-like open-ocean processes and water-mass mixing to be established, which decreased the sensitivity of the Nd-isotope composition of seawater to local effects such as terrigenous and coastal sedimentary inputs. The convergence of Nd-isotope signatures across the entire Atlantic Ocean spanning paleo-

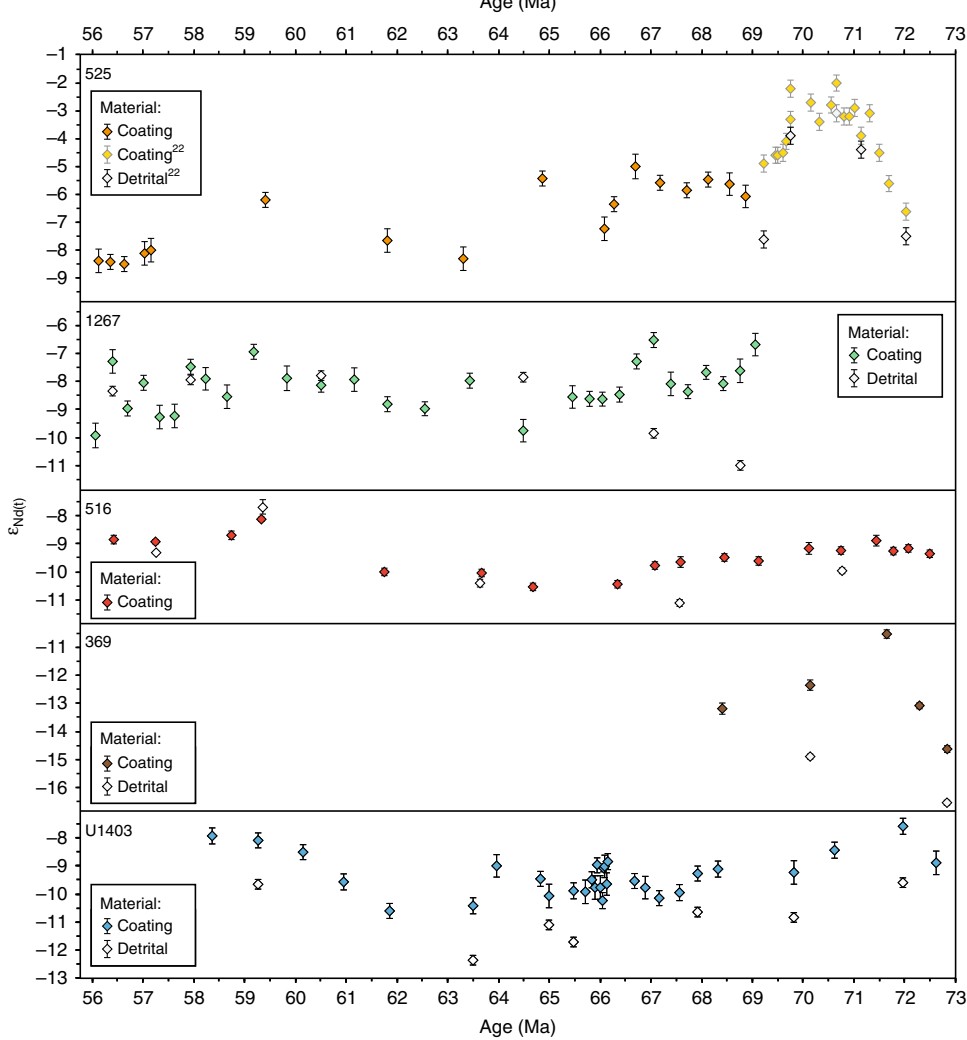

**Fig. 3** New Nd-isotope results. Neodymium isotope values of ferromanganese coatings (coloured symbols) and detrital fractions (open symbols) measured in this study and previously for Site 525[22]. Error bars indicate 2 s.d. external reproducibility

waterdepths of 500 to 4500 m, further suggests that between 62 and 59 Ma, both local tectonic restrictions as well as the vertical stratification of the deep Atlantic Ocean decreased and a global mode of thermohaline circulation was initiated.

The close correspondence in Nd-isotope values among sites at 59 Ma coincided with the onset of the mid-Paleocene global climate warming as evident from benthic foraminiferal oxygen isotopes[13,41] (Fig. 4). Based on a recent reconstruction of continental rift length histories[11] in comparison to the long-term evolution of atmospheric $pCO_2$[8], the underlying cause of this warming may lie in the increased cumulative length of incipient continental rifts. Despite a reconstructed gradual increase in $pCO_2$ levels during the end of the Cretaceous and earliest Paleocene[8,11] (Fig. 4), as well as an initial magmatic phase of the North Atlantic Igneous Province from 62 to 61 Ma[13], the long-term increasing trend in bottom-water temperatures did not start until 59 Ma[13,41], when $pCO_2$ started to increase at a higher rate[8,11] (Fig. 4).

## Discussion

We hypothesize that the strengthened Atlantic overturning circulation suggested by our data enhanced oceanic poleward heat transport thereby contributing to global climate warming culminating in the peak greenhouse conditions of the Eocene. Global

warming may itself have enhanced vertical mixing through increased occurrence of storms and cyclones[42] that enabled more efficient overturning circulation in the Atlantic Ocean. Both the deepening of the Rio Grande Rise and enhanced mixing associated with global warming would have increased the capacity of the overturning circulation in the Atlantic Ocean to transport heat. These interpretations of our new Nd-isotope data are consistent with observed changes in Late Cretaceous to early Paleogene Nd-isotope records from the Pacific Ocean[42] and Earth system modelling results, which indicate that vigorous ocean circulation and strong vertical mixing resulted in increased oceanic heat transport and reduced equator–pole temperature gradients[42,43]. Higher oceanic heat transport efficiency likely also set the stage for the occurrence of brief hyperthermals which were frequently superimposed on the overall temperature rise of the Eocene hothouse[41]. Together with increasing atmospheric $CO_2$ levels[8,11], the changing paleogeography of the Atlantic Ocean may have contributed to the boundary conditions that pushed the Earth's climate into a greenhouse state.

## Methods

**Extraction of Nd isotopes**. For Nd-isotope analyses of past seawater extracted from ferromanganese oxide coatings, bulk sediment samples consisting mainly of nannofossil oozes and chalks were dried and homogenised in an agate mortar. To extract the authigenic, seawater-derived Nd-isotope signature, ~2.5 g of powder

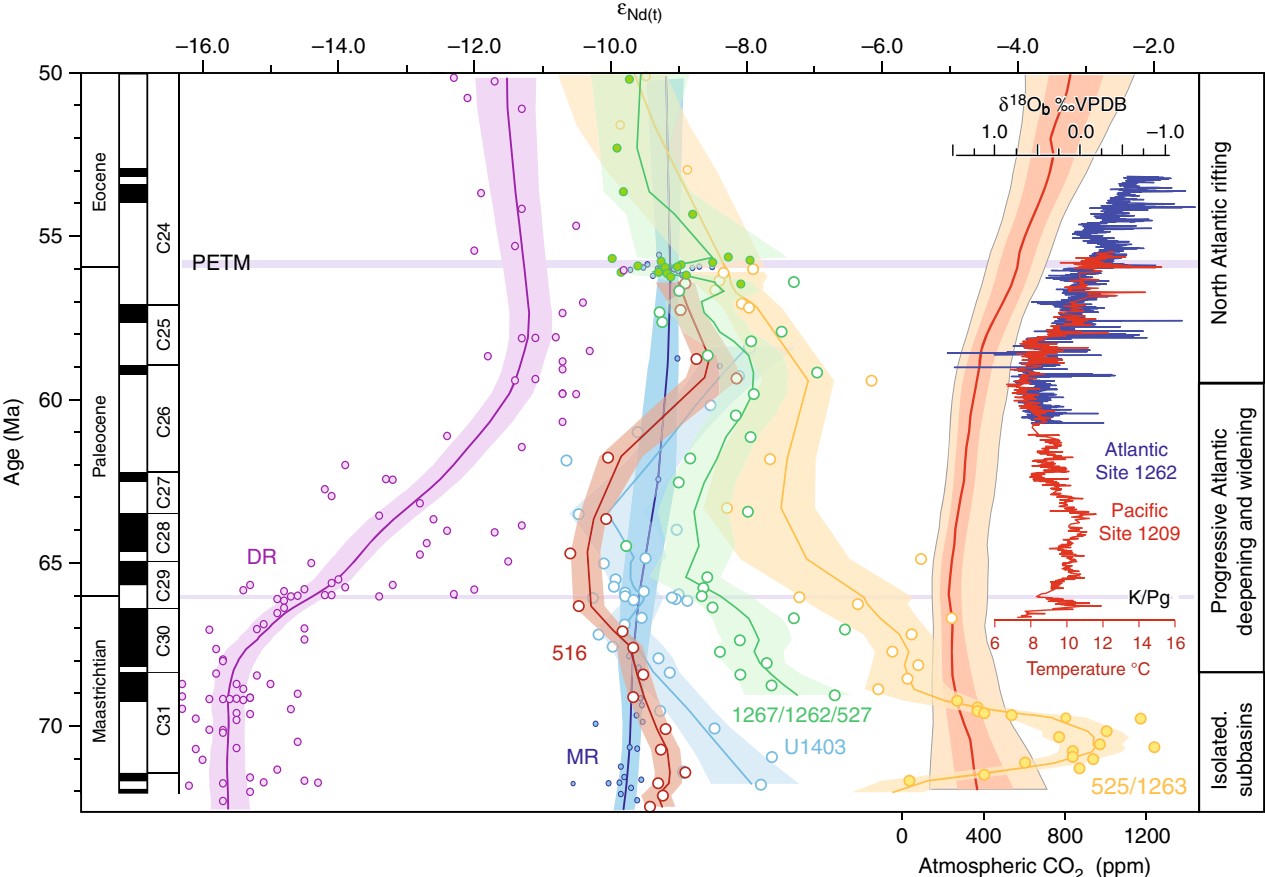

**Fig. 4** The convergence of Nd-isotope values. Maastrichtian to early Eocene Nd-isotope records compared to atmospheric $CO_2$ levels and deep-sea temperature evolution. Open symbols depict new $\varepsilon_{Nd(t)}$ data for North Atlantic Site U1403 (Newfoundland margin, dark blue) and South Atlantic Sites 516 (Rio Grande Rise, red), 1267 (northern flank of the Walvis Ridge, green) and 525 (top of the Walvis Ridge, yellow). Filled symbols depict selected high-resolution $\varepsilon_{Nd(t)}$ data from Demerara Rise in the North Atlantic[26,28] (purple), Sites 527[33] and 1262[34] on the flank of the Walvis Ridge (green), Site 1263[34] and Site 525[22] on the Walvis Ridge (yellow), and Maud Rise in the Southern Ocean[20] (dark blue). The $\varepsilon_{Nd(t)}$ data have been smoothed by LOWESS regression ($f = 0.3$) with the shaded areas representing the 97.5% confidence interval. The $\varepsilon_{Nd(t)}$ data are flanked by the geomagnetic polarity time scale on the left and on the right by reconstructed atmospheric $CO_2$ levels[11], benthic oxygen isotope data from the Pacific[41] and the Atlantic[13], and the suggested evolution of the Atlantic Ocean. The Paleogene age model of the Sites rests on a combination of magnetic and nannofossil stratigraphy plotted against the GTS 2012[64]. C31-C24: paleomagnetic chrons, $\delta^{18}O_b$: oxygen isotope ratio from benthic foraminifera, VPDB: Vienna Pee Dee Belemnite standard, PETM: Paleocene/Eocene Thermal Maximum, DR: Demerara Rise, K/Pg: Cretaceous/Paleogene boundary, MR: Maud Rise

was treated following the procedure described in ref. [44], omitting the carbonate removal step[45]. Powdered samples were rinsed three times with de-ionized (MQ) water, after which 10 ml of MQ was added and 10 ml of a 0.05 M hydroxylamine hydrochloride/15% acetic acid solution, buffered with NaOH to a pH of 4. Samples were placed on a shaker for 1 h and centrifuged. The supernatant containing the seawater Nd-isotope signature of the ferromanganese oxide coatings was pipetted off and dried down. For determining the detrital $\varepsilon_{Nd}$ signature, selected samples underwent an additional 12 h leaching step with 20 ml of the hydroxylamine solution (above), after which samples were rinsed with MQ three times and ~50 mg of dried sample was dissolved in a mixture of aqua regia and HF. As preparatory steps for column chemistry, all samples were refluxed in concentrated $HNO_3$ at 80 °C overnight, centrifuged, and 80% of the supernatant was dried down. Twice, 0.5 ml of 1 M HCl was added and the sample was dried down, after which the samples were redissolved in 0.5 ml 1 M HCl. Samples were passed through cation-exchange columns with 0.8 ml AG50W-X12 resin (mesh size 200-400 μm), using standard procedures, to separate Sr and the Rare Earth Elements (REEs), as well as removing most of the Ba[46]. A second set of columns with 2 ml Ln-Spec resin (mesh size 50-100 μm) was used to separate Nd from the other REEs and remaining Ba[47].

**Analytical procedure.** Neodymium isotope ratios were measured on a Nu Instruments Multiple Collector Inductively Coupled Plasma Mass Spectrometer (MC-ICPMS). The majority of samples were measured at GEOMAR Kiel, Germany, and a subset of samples at the department of Earth Sciences of Oxford University, UK (Supplementary Tables 2, 5 and 6). Measured $^{143}Nd/^{144}Nd$ results were mass-bias corrected to a $^{146}Nd/^{144}Nd$ ratio of 0.7219 and were normalized to

the accepted $^{143}Nd/^{144}Nd$ value of 0.512115 for the JNdi-1 standard[48], which was measured after every third sample.

The results were decay-corrected for the time of deposition by $(^{143}Nd/^{144}Nd)_{sample(t)} = (^{143}Nd/^{144}Nd)_{sample(0)} - [(^{147}Sm/^{144}Nd)_{sample(0)} * (e^{\lambda t} - 1)]$ where $t$ is time, the decay constant $\lambda$ is $6.54 \times 10^{-12}$, and using an average $^{147}Sm/^{144}Nd$ ratio of 0.124[22]. Nd-isotope ratios are reported as $\varepsilon_{Nd(t)}$ values with respect to the Chondritic Uniform Reservoir (CHUR), which are calculated as $\varepsilon_{Nd(t)} = [(^{143}Nd/^{144}Nd)_{sample(t)} / (^{143}Nd/^{144}Nd)_{CHUR(t)} - 1] \times 10^4$ using a $(^{143}Nd/^{144}Nd)_{CHUR(0)}$ value of 0.512638, and a $(^{147}Sm/^{144}Nd)_{CHUR(0)}$ of 0.1966[49]. External reproducibility ($2\sigma$) of the measurements was between 0.15 and 0.54 $\varepsilon_{Nd}$ units and procedural Nd blanks were ≤ 30 pg Nd and thus negligible.

**Age models.** Age models for the individual sites were generated by an integrated approach of magneto- and biostratigraphy and if available astrochronology. All datum levels are tied to the Geological Timescale GTS2012. Ages of polarity chrons are from ref. [50], and of calcareous nannofossils (NP zonation) from ref. [51] as compiled in ref. [52]. In detail the following data are used and summarized in Supplementary Table 7. Tie points for Site 516 are defined by magneto- and calcareous nannofossil stratigraphy given in ref. [53]. Tie points for Site 525 are defined by polarity chrons in the Maastrichtian[54] and by calcareous nannofossils in the Paleocene[55]. Tie points for Site 1267 are derived from precession cycle counting for the upper Paleocene (until 58.2 Ma ago)[56] and polarity chrons for the lower to middle Paleocene and the Maastrichtian[57]. Ages of neodymium isotope data from Site 1262[34] and Site 527[33] were converted to GTS 2012. Tie points for Site 369 follow the age model of the Shipboard Scientific Party[58]. Tie points for Site U1403 are defined by first occurrences (FO) of calcareous nannofossils for the Paleocene[59]

with an adjustment for the FO of *Lithoptychius* spp. at 227 m depth rCCSF (corresponding to the first radiation of fasciculithids according to refs. [60],[61]) and by astronomical tuning of 405 kyr cycles and carbon isotope stratigraphy[62].

## Data availability

The authors declare that all the data generated during this study are available within the manuscript and its supplementary information file.

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

## Acknowledgements

We would like to thank the International Ocean Discovery Program (IODP) for providing samples. Authors were funded by the German Research Foundation (DFG) under grant numbers DFG VO 687/14, FR2544/8, and FR1198/11, BO2505/8 and EH89/20. We would like to thank Jutta Heinze and Chris Siebert at GEOMAR, Kiel and Alan Hsieh at Oxford University, UK for smooth operation of the laboratory and the mass spectrometers.

## Author contributions

S.J.B., S.V., O.F. and M.F. developed the project. S.J.B., A.H.O. and T.K. performed Nd-isotope analyses. L.P-D. provided bathymetric information. All authors contributed to the writing of the manuscript.

## Additional information

**Competing interests:** The authors declare no competing interests.

