## [Peer Review File · Nature Communications]

Reviewer #1 (Remarks to the Author):

This manuscript presents neodymium isotopic ratios from Late Cretaceous and Paleogene Atlantic deep sea sites. It suggests there is a large spread in ϵ_{Nd} values among sites in older samples with values converging up section. The paper argues this convergence indicates a “major intensification of deep water exchange” at the time temperatures began to increase in the Paleocene as inferred from benthic foraminiferal d_{18O} values. The paper contends that subsiding barriers to deep water flow may be the ultimate cause of circulation changes and that the inferred reorganization of circulation forced warming (although increasing CO_2 levels are also mentioned as a potential cause). The data seem to be of high quality and are relevant, the topic is of interest to a widespread audience, the writing is clear, and the study seems worthy of publication. However, moderate revisions are suggested. In addition to a number of specific comments below, the following larger conceptual and/or theoretical claims need to be more completely and/or explicitly addressed.

The interval studied is proposed to span the end of a time of cooling and the beginning of a time of warming yet Fig. 1 and the text suggest progression convergence of neodymium ratios among sites across the entire interval. If convergence from 64 to 59 Ma leads to warming, why does convergence between 70 and 65 Ma correlate with cooling?

Data from the North Atlantic are sparse, the pattern at U1403 seems to start at the ‘converged’ value, drifts lower from 70-62 Ma before drifting higher from 61 to 58 Ma (thus, it could be argued it is the low values that are the excursion or important change in the North Atlantic). Including more North Atlantic data from supplementary figure 1 would not obviously help support the claims in the text.

Along similar lines, data in Figure 1 for the South Atlantic are from a limited region, and it is not clear whether plotting the more widespread data in Supp. Figure 1 would support the asserted pattern. Also, how does this Paleogene pattern relate to Nd convergence argued to occur earlier in the Late Cretaceous and interpreted to be the initiation of the formation of early Antarctic deep waters?

How do changes in neodymium ratios indicate how vigorous circulation was (line 124)? Connectivity seems an intuitive inference from the observation of similar values, but how are rates inferred?

Finally, does change in circulation cause global warming or is it more likely to affect regional distribution of warmth? As noted below, lines 134 and 135 suggest the former while line 139 concludes the latter. This distinction seems critical to the major implications of the paper and needs more discussion and clarity.

line by line comments:

17: Late

17-19: refs. 1 & 3 may not be best to support claims made

41-42: refs. may not be best to support claims, and statement implies consensus that may not exist.

44: citation needed

66: does ref. 11 include Nd data?

73: not clear how Fig. S1 illustrates this point

77: figure shows 4 regions and lists 7 sites; neither match the 5 sites claimed

82: analytical methods for detrital fraction do not seem to be presented

89: 'This' unclear what pronoun replaces

92: claim that sites are from 'far enough' offshore settings that measurements capture water mass values seems to contradict preceding claim that similarity between detrital and leachate values indicates importance of detrital input

102: is Atlantic 1 basin, separate North and South Atlantic basins, or multiple basins? Presentation seems to shift among these possibilities

103: are there large differences in maximum depth and basin width between 70 Ma and 60 Ma especially relative to changes across the history of the basin? If not, what makes this interval a turning point?

113-115: depths are cited in several places and paleodepths are given for sites, but there does not seem to be discussion of possible depth related differences in neodymium isotopes among sites. Are there any?

117: North Atlantic data seems to end at 58 Ma (U1403) whereas data from younger than 59 Ma seems to be from one area in the South Atlantic especially for points younger than 56 Ma. Thus, claims that basin-wide patterns are documented seems overstated.

119: 'This' unclear what pronoun replaces

123: see 117

134-135: does change in circulation cause global warming or does it affect regional distribution of warmth. This distinction seems critical and needs more discussion. Line 139 suggests latitudinal not global changes are the result of circulation changes.

140-143: arguments beg for expanded presentation

160: were Sm/Nd ratios measured in samples?

266-267: move parenthetical comment so that it immediately follows 'North Atlantic Ocean.' Overall caption was difficult to interpret

Supplementary material:

discussion of detrital analyses is minimal

citation call outs are confusing

there seems to be a difference in statement criteria for including a site (5 pts. in text, 4 pts. in SM).

not clear data in SF1 support claims made in text, lines and errors envelopes (?) not described, Cape Verde typo in legend, are all data from leached samples?

table footnotes are not clear... e.g., 'of "TD" indicate...?', 'samples were combined'?

Reviewer #2 (Remarks to the Author):

This is the review of the paper written by Batenburg et al. and submitted to Nature Communications. I must first apologize for having retained the manuscript because of my late review. I also have to clearly state that my main domain of expertise is on climate modeling. Likewise, my review should be seen as one from a non-specialist of Nd. The paper focuses on Nd isotope records over the 72-56 Ma period from Rio Grande Rise, Walvis Ridge (South Atlantic) and Newfoundland basin (North). Measurements have been made on Fe-Mn coatings of sediments and on detrital fractions when it was possible. This technic should allow in principle to disentangle the oceanographic from the continental source signal as Nd isotopes record both signal. Whatever the interpretations the authors have of the similar trend followed by both coatings and detrital fraction, the 4 IODP coatings records in Figure 2 show a convergence toward values comprised between -10 and -8. The convergence trend starts at 65-63 Ma and is complete at 59-58 Ma. The authors use these synchronous trends to propose the onset of more intense deep-water circulation in the Atlantic basin, which in turn, should explain the fact that Nd isotope values become closer whatever their geographical distribution were in the Atlantic Ocean. Although I do agree with these conclusions, the authors go a step further and discuss the potential implications of a well-mixed Atlantic ocean. At this time, the paper becomes more hypothetical. Based on a study of Thomas et al. (2014), the authors suggest that a more ventilated deep ocean (less restricted) may increase the ocean heat transport which in turn would feedback on the thermal latitudinal gradient and on the global climate (and temperature). They then suggest that the early warming phase of the Paleogene may be ascribed to the intensification of deep-water circulation.

In the paper of Thomas et al. (2014), using early Eocene boundary conditions, the early Cenozoic circulation simulated by the MITgcm is rather sluggish and it is only by artificially increasing the

vertical mixing coefficient by 50% (potentially attributable to changes in tidal mixing) and then by 500 % at every depth (very unlikely or at least unconstrained) that Thomas et al. are able to simulate an intense oceanic circulation in the Pacific Ocean. So, in principle, it is not the changing paleogeography that induces better-ventilated water in the Thomas et al.'s paper but rather some other processes. For a small part, tidal mixing, for a larger part, severe storms and tropical cyclones hypothesized to occurred more frequently in a warmer world may have provided the energy required by the ocean for a global increase in its mixing capability not the evolution of the paleogeography. In that sense, it is the global warming occurring during the early Paleogene that may have induced larger mixing rate in the ocean at that time.

While the authors do focus on the convergence trend in their Nd records, the common increasing and decreasing trends have not been under scrutiny (Fig. 2). A decrease from 70 to 65 Ma then an increase from 65 to 59 Ma and then again a decrease from 59 to 50 Ma are clearly visible. I don't know if it is meaningful but this may be of interest if explanations exist.

In general, I find the paper very well written but too short in terms of explanations or at least uneasy to follow because many hypotheses are done. As there is room for figures and text in Nature Communications, I suggest to the authors to bring back the 2 supplementary figures in the main text and in general to extend discussions whenever it is possible. This paper represents a strong analytical work with many measurements but it fails to be fully informative because many results have been left aside. A good synthesis of what has been done till now is lacking. In the Figure 1, you show sites on which Nd data are existing and sites for which new data have been acquired (U1403, 369, 1267, 525). The link between this figure and the next one is not clear at all. On Figure 2, you are now referring to Rio Grande (516F), Walvis Ridge (two sites, 525 and 1267) and Newfoundland basin (1403). So, site 369 has disappeared and is now site 516F, meaning that you have 3 South Atlantic sites very close to each other and only one in the North Atlantic. This makes all your interpretations in term of deep-water masses less convincing. In addition, why all data already existing are not plotted on Figure 2 with your new data? It is not clear to me. Because, you are showing all pre-existing data on Figure 1 (all orange circles), it is important to plot their Nd values against your own data? This could make your argument for a homogenous deep-water masse expanding in the whole Atlantic Ocean more convincing.

As a consequence, I would like to see eventually this article published in Nature Communications but in the present state, I don't think that it crosses the threshold to be published. My recommendation is to clarify all various points putted forward in this review, not simply in the response to reviewers, but directly within the main text, in order to make this paper more meaningful, accounting for previous works as much as possible and providing a clear statement of science advances get through new data presented here.

Below are my specific comments:

Abstract:

Ref. 2 and 3 are not appropriate, the latter being on Cenozoic CO₂ reconstruction and not on Late Cretaceous, and the former being a bit at odd with what has been achieved today. I would advice the paper by Brune et al. (2017, Nature Geoscience) that makes a strong point on correlating the decreasing length of continental rifting with atmospheric CO₂ level during the Late Cretaceous. Another study (Pucéat et al., 2004, Geology) showed that the Late Cretaceous cooling was occurring uniformly at the latitudes 10-50° strongly suggesting a general decrease of the atmospheric CO₂ level as an explanation rather than a reorganization of the ocean-atmosphere dynamics.

Lines 38. You should consider adding the Nature paper by Gutjahr, A., Ridgwell an co-authors in Nature, 2017

Lines 66. Ref should be 11-17?

Lines 64-66. The figure S1 without detailed explanations is hard to read. Please provide more details in the text and not simply in the legend. You also give 6 references here but it is hard to see what we need to keep as a message? > It is complicated and there are many hypotheses.

In addition, in the following sentences, you suggest that Nd signatures fall within a narrow range of -10 to -8 in the Atlantic basin. Your study thus confirms this narrow range but allows going more away back in time? Am I right? It is something I had trouble to decide. Is your study providing something new in terms of understanding of the evolution of the global circulation? Or is your study confirming other results? It is probably because you have condensate the manuscript a lot that I am not really able to deal with this question.

Lines 71-73. Figure S1 show more Nd data, I don't see the point made here concerning the enlarging of the Atlantic Ocean and the modern THC-like? Is there a link between Fig. S1 and all references cited just before?

Lines 84-94. This is where you state that Nd data do record an oceanographic signal and not a continental-weathering signal. I am not sure to get your point here? How is it possible to conclude that your record reproduces the oceanographic signal if both detrital and coatings follow the same trend? From Figure S2, several long-term trends in Nd are visible on both records although

significantly offset as you do recognize. In addition, your argument that offshore records far enough from the coast cannot be influenced by detrital inputs is plausible but then why the detrital record follows the coating's one?

After a discussion with K. Tachikawa and E. Puceat, I think you should consider extending the discussion here to suggest that authigenic sediment may explain why detrital records follow the coating's one (see Moiroud et al., 2016 and Tachikawa et al., 2016)

Lines 124-127. Do you see an increased difference between detrital and coatings values across the time period you are working on? If not, this sentence is not supported by your own data.

Lines 132-134. I don't agree, the convergence starts earlier while temperature were cooling from 65 to 59 Ma

Lines 142. "Together with increasing atmospheric CO₂ levels³, the changing paleogeography of the Atlantic thus likely contributed significantly to the boundary conditions that pushed the Earth's climate into a greenhouse state."

The paper, ref.3 you are referring to, suggest that atmospheric CO₂ levels decreased while the climate was cooling from 54 to 38 Ma. I am sure there are other papers that CO₂-proxies showed an increase at this time. This reference is not the right one.

Figure 2: if Rio Grande Rise is so close to Walvis Ridge, why so much differences in Nd Maastrichtian values? What is the history of depth of these sites within the time interval ? Is there a possibility that you are also recording the deepening of your IODP sites ?

Reply to Reviewers' comments

Dear reviewers,

Thank you very much for reviewing our manuscript. We are grateful for the careful evaluation of the data and the research, and for your comments and suggestions that helped us to improve the manuscript and our thinking about the oceanographic and climatic processes involved.

This document provides a detailed reply to each comment. Below, we have copied the reviewers' comments in black and added our replies in blue.

To summarize we have:

- moved the figures from the supplementary information to the main manuscript and revised the map and the synthesis figure to include more data;
- included an improved discussion of the Nd-isotope results of the detrital fraction;
- discussed the decreasing and increasing trends in our Nd isotope data and potential causes in more detail;
- refined our argument on the potential link between enhanced circulation and heat exchange in relation to atmospheric CO₂ levels.

In addition, we have invited Lucia Pérez-Díaz to join as a co-author, to help evaluate the role of changes in bathymetry in more detail and we have included a 3D reconstruction of paleobathymetry in Figure 2.

We hope that the changes address all issues raised and hope that our manuscript is now acceptable for publication. Thank you for your insightful reviews.

Kind regards, on behalf of all authors,

Sietske Batenburg

Reviewers' comments:

Reviewer #1 (Remarks to the Author):

This manuscript present neodymium isotopic ratios from Late Cretaceous and Paleogene Atlantic deep sea sites. It suggests there is a large spread in eNd values among sites in older samples with values converging up section. The paper argues this convergence indicates a "major intensification of deep water exchange" at the time temperatures began to increase in the Paleocene as inferred from benthic foraminiferal d18O values. The paper contends that subsiding barriers to deep water flow may be the ultimate cause of circulation changes and that the inferred reorganization of circulation forced warming (although increasing CO₂ levels are also mentioned as a potential cause). The data seem to be of high quality and are relevant, the topic is of interest to a widespread audience, the writing is clear, and the study seems worthy of publication. However, moderate revisions are suggested. In addition to a number of specific comments below, the following larger conceptual and/or

theoretical claims need to be more completely and/or explicitly addressed.

1)

The interval studied is proposed to span the end of a time of cooling and the beginning of a time of warming yet Fig. 1 and the text suggest progression convergence of neodymium ratios among sites across the entire interval. If convergence from 64 to 59 Ma leads to warming, why does convergence between 70 and 65 Ma correlate with cooling?

Thank you for identifying concepts that required better explanations in the text. We did not clearly distinguish between the process of converging Nd-isotope ratios and the values having converged. We have now adjusted the wording in lines 154 and 171.

We interpret the close similarity of Nd-isotope signatures at 59 Ma to be indicative of the surpassing of a threshold, namely that the Rio Grande Rise subsided sufficiently to allow for significant volumetric exchange of deep waters between sub-basins of the opening Atlantic Ocean. Prior to 59 Ma, local inputs and processes such as boundary exchange may have had a strong influence on the Nd-isotope signature of the seawater. Such influences may have decreased through time as volumetric flow increased.

Related to your last point (5) below, a change in the efficiency of circulation may have contributed to the distribution of heat over the planet rather than being the underlying cause of warming. We now discuss the convergence in Nd-isotope values in relation to the evolution of atmospheric pCO₂ and continental rifting (lines 171-194).

2)

Data from the North Atlantic are sparse, the pattern at U1403 seems to start at the 'converged' value, drifts lower from 70-62 Ma before drifting higher from 61 to 58 Ma (thus, it could be argued it is the low values that are the excursion or important change in the North Atlantic). Including more North Atlantic data from supplementary figure 1 would not obviously help support the claims in the text.

We agree with these observations and would like to emphasize that it is exactly the point of our interpretation that prior to the convergence of the signatures and trends in Nd isotope signatures, the signatures of different sites were more dominated by either local weathering inputs or local water mass mixing within the sub-basins and then became similar due to the improved connectivity of the sub-basins. The data from Demerara Rise, the highest resolution Nd-isotope data set available for any region in the North Atlantic, are now incorporated in Figure 4 (originally Figure 2). These data remain offset from the new data presented in this manuscript due to the persistent influence of unradiogenic local weathering inputs but, importantly, show a similar converging trend from 65 to 59 Ma, after which the evolution of the Nd isotope signature remained parallel to the Atlantic datasets presented here. The data from Demerara Rise have now been incorporated in the discussion at lines 141-143.

For U1403, the negative trend into the Paleocene and the subsequent positive trend up to ~59 Ma are discussed in more detail now in relation to deepening of the studied sites and potential changes in the balance between radiogenic and unradiogenic sources of Nd (lines 125-145).

3)

Along similar lines, data in Figure 1 for the South Atlantic are from a limited region, and it is not clear whether plotting the more widespread data in Supp. Figure 1 would support the asserted pattern. Also, how does this Paleogene pattern relate to Nd convergence argued to occur earlier in the Late Cretaceous and interpreted to be the initiation of the formation of early Antarctic deep waters?

We have now included former Figure S1 as the first figure in the main manuscript, and the highest resolution data records of the Paleocene, from Demerara Rise and Maud Rise, are now included in our synthesis Figure 4 (originally Figure 2).

The Paleocene convergence in Nd-isotope values does indeed resemble parallel ϵ_{Nd} trends between the North and South Atlantic that have been observed earlier in the Late Cretaceous (~83 to 78 Ma), and that are interpreted to reflect the formation of a southern-sourced deep water mass, “Southern Component Water”, that made its way through the Atlantic Ocean (Robinson and Vance, 2012). Although these older ϵ_{Nd} trends were apparently parallel over tens of millions of years, the time series at the different sites show distinct and dissimilar variability, and the spread in values is large: approximately 5 ϵ_{Nd} units between 77 and 75 Ma, excluding the more negative values of Demerara Rise of -14 to -16 (MacLeod et al., 2011; Martin et al., 2012; Robinson et al., 2010; Robinson and Vance, 2012). In contrast, our new Paleocene ϵ_{Nd} data from Sites 525, 1267 and 516 in the South Atlantic, and Site U1403 in the North Atlantic do not only display parallel trends on a finer time scale of millions of years, but also a convergence of absolute values at 59 Ma, with all ϵ_{Nd} values falling between -7 and -9.5 for the period 58.5–56.5 Ma, when only the ϵ_{Nd} values from Demerara Rise averaged -11 (MacLeod et al., 2011). We propose that the decreasing isotopic trend earlier in the Late Cretaceous reflected the ongoing separation of the South American and African continents. This allowed for a surface-to-intermediate water connection across the equatorial gateway and increased connectivity between the North and South Atlantic as well as increased surface connectivity with the Southern Ocean. The converging Nd-isotope values at 59 Ma likely reflect the point in time at which the Rio Grande Rise – Walvis Ridge barrier had sufficiently subsided to allow an efficient exchange of deep waters.

The parallel trends earlier in the Late Cretaceous are discussed in lines 83-87, and the trends observed in our datasets, are now discussed in more detail in lines 125-145.

4)

How do changes in neodymium ratios indicate how vigorous circulation was (line 124)? Connectivity seems an intuitive inference from the observation of similar values, but how are rates inferred?.

The close correspondence of Nd-isotope values at 59 Ma suggests that essentially one water mass was able to make its way across the Atlantic Ocean without its signature being affected by overprinting through regional inputs or boundary exchange mechanisms. The latter processes influenced the Nd-isotope signatures at the different sites to a decreasing extent up to 59 Ma. We suggest that the simplest process to result in the convergence of Nd-isotope values, under the assumption that Nd inputs through weathering and boundary exchange did not suddenly decrease, would be an increase in the volume flow of a deep-water mass passing over the Rio Grande Rise and making its way northward. Given that we do not know the amounts of past weathering or boundary exchange fluxes and that the Nd

isotope method does not allow determining past concentrations of Nd in seawater, we cannot calculate the required volumetric exchange and the inference of more vigorous circulation at 59 Ma thus remains a qualitative statement. We have deleted “more vigorous” and now refer to “increased volumetric exchange” (line 163).

5)

Finally, does change in circulation cause global warming or is it more likely to affect regional distribution of warmth? As noted below, lines 134 and 135 suggest the former while line 139 concludes the latter. This distinction seems critical to the major implications of the paper and needs more discussion and clarity.

As Reviewer 1 suggests, the change in global circulation patterns did not cause warming but rather affected the distribution of heat. Potential causes of warming are increased continental rifting and/or the first phase of magmatic activity of the North Atlantic Igneous Province, prior to break-up leading to enhanced release of CO₂ to the atmosphere. The discussion on the relationship between global warming, increased deep-water exchange and heat transport has been expanded in lines 171 to 194, which includes: “We hypothesize that the strengthened Atlantic overturning circulation suggested by our data enhanced oceanic poleward heat transport thereby contributing to global climate warming culminating in the peak greenhouse conditions of the Eocene.” (lines 180-182).

line by line comments:

17: Late

Corrected

17-19: refs. 1 & 3 may not be best to support claims made

This has been adjusted and expanded to Clarke and Jenkyns, 1999; Cramer et al., 2009; Friedrich et al., 2012; Huber et al., 2018, 1995. These references have now been included in the first sentence of the introduction (lines 32-33), rather than in the summary paragraph.

41-42: refs. may not be best to support claims, and statement implies consensus that may not exist.

The references, as well as the statement have been adjusted to: “the ongoing opening and deepening of the Atlantic basin (Pérez-Díaz and Eagles, 2017; Sewall et al., 2007) led to increased North-South connectivity, although the timing of the establishment of a deep-water connection remains debated (Donnadieu et al., 2016; Friedrich et al., 2012; Moiroud et al., 2015; Robinson et al., 2010; Voigt et al., 2013).” (lines 44-47).

44: citation needed

This is adjusted by adding “likely”, to reflect that this is our own statement (line 47).

66: does ref. 11 include Nd data?

Thank you for noticing, this reference has been removed.

73: not clear how Fig. S1 illustrates this point

The reference to the figure has been deleted and a reference to Donnadiou et al. (2016) has been added (line 98).

77: figure shows 4 regions and lists 7 sites; neither match the 5 sites claimed
Former Figure 1, now Figure 2, has been adjusted and color coded in agreement with the symbols in former Figure 2, now Figure 4.

82: analytical methods for detrital fraction do not seem to be presented
Lines 201-203 in the main manuscript listed the dissolution procedure, but failed to specify that exactly the same column chemistry protocol as for the ferromanganese oxide coatings was followed afterwards. This has now been expanded accordingly in lines 379-381 of the supplementary information.

89: 'This' unclear what pronoun replaces
We have combined and shortened the statement, omitting the word "This" (line 115).

92: claim that sites are from 'far enough' offshore settings that measurements capture water mass values seems to contradict preceding claim that similarity between detrital and leachate values indicates importance of detrital input

We agree that this was a confusing statement. We were aiming to express that the ferromanganese oxide coatings faithfully recorded the seawater Nd-isotope signature, but that the seawater signature itself were likely strongly influenced by boundary exchange processes. We have adjusted this section (lines 109-117) to clarify this issue and to include suggestions raised by Reviewer 2 (see below).

102: is Atlantic 1 basin, separate North and South Atlantic basins, or multiple basins?

Presentation seems to shift among these possibilities

We have replaced "basin" by "basins" in this sentence, and throughout the manuscript "Ocean" has been added after "Atlantic".

103: are there large differences in maximum depth and basin width between 70 Ma and 60 Ma especially relative to changes across the history of the basin? If not, what makes this interval a turning point?

At 60 Ma reconstructed bathymetries (Pérez-Díaz and Eagles, 2017) estimate the western portion of the Rio Grande Rise to have subsided below 2500 m water depth, whereas the Rio Grande Rise was above 2000 m at 70 Ma with the exception of the Vema and Hunter Channels. We infer that the efficient exchange of deep water was only possible once a larger portion of the Rio Grande Rise-Walvis Ridge barrier had subsided, in this case the westernmost portion of the Rio Grande Rise according to the bathymetry reconstruction of Pérez-Díaz and Eagles (2017). We have included a three-dimensional plot of the South Atlantic bathymetry in Figure 2.

113-115: depths are cited in several places and paleodepths are given for sites, but there does not seem to be discussion of possible depth related differences in neodymium isotopes among sites. Are there any?

The most striking difference in ϵ_{Nd} data that may be related to depth is the offset between Nd-isotope ratios of Sites 525 and 1267 in the latest Cretaceous. Site 525 was located at a

relatively shallow water depth of ~1300 m, whereas Site 1267 was located near the base of the northwestern slope of the Walvis Ridge at ~3000 m. The most positive values of Site 525 have been interpreted to reflect an episode of volcanic activity on the Walvis Ridge itself (Voigt et al., 2013). The fact that Nd-isotope ratios at Site 525 remain positively offset from those at Site 1267 indicates that the shallow water mass bathing Site 525 remained influenced by relatively radiogenic inputs of Nd until the end of the Cretaceous, possibly by weathering inputs from the Walvis Ridge itself, which was partially subaerially exposed (Pérez-Díaz and Eagles, 2017). The offset in ϵ_{Nd} values between Sites 525 and 1267 gradually decreased in the Paleocene and we interpret this as a consequence of both the deepening of the studied sites and the increased efficiency with which one deep-to-intermediate water mass was able to be advected across the Atlantic Ocean without its Nd-isotope signature being overprinted by local processes, similar to the N-S flow of North Atlantic Deep Water in the present day North Atlantic. We have added a discussion on the depths and the deepening of the sites at lines 142–157.

117: North Atlantic data seems to end at 58 Ma (U1403) whereas data from younger than 59 Ma seems to be from one area in the South Atlantic especially for points younger than 56 Ma. Thus, claims that basin-wide patterns are documented seems overstated.

We have added “at 59 Ma” (line 154), as our own records do not span beyond 58 Ma for the North Atlantic and beyond 56 Ma for the South Atlantic. Records from other localities are generally of limited resolution beyond 56 Ma, but our results from Site U1403 at 58 Ma are corroborated by three data points measured at Site 549 (Goban Spur) that vary between -8 and -9.2 at 57 Ma. This is now included in the text at lines 122-124.

119: ‘This’ unclear what pronoun replaces

“This” is replaced by “Such a southern origin of deep water” (line 158).

123: see 117

We hope to have addressed this with the addition of text as described above.

134-135: does change in circulation cause global warming or does it affect regional distribution of warmth. This distinction seems critical and needs more discussion. Line 139 suggests latitudinal not global changes are the result of circulation changes.

To better distinguish between the source of warming and the contribution of ocean circulation to the distribution of heat, we have expanded the discussion in lines 173 to 194, including the following text: “Both the deepening of the Rio Grande Rise and enhanced mixing associated with global warming would have increased the capacity of the overturning circulation in the Atlantic Ocean to transport heat. These interpretations of our new Nd isotope data are consistent with observed changes in Late Cretaceous to early Paleogene Nd isotope records from the Pacific (Thomas et al., 2014) and Earth system modelling results, which indicate that vigorous ocean circulation and strong vertical mixing resulted in increased oceanic heat transport and reduced equator–pole temperature gradients (Sijp and England, 2016; Thomas et al., 2014).”

140-143: arguments beg for expanded presentation

We hope to have addressed this by addition of the text in lines 173-194, which discusses the relationship between CO₂ increase, circulation and heat transport in the development of greenhouse conditions in the Eocene (line 182).

160: were Sm/Nd ratios measured in samples?

No, we have used a published ¹⁴⁷Sm/¹⁴⁴Nd ratio of 0.124 (Voigt et al., 2013). Small variations in this ratio would not alter the results to a significant extent.

266-267: move parenthetical comment so that it immediately follows 'North Atlantic Ocean.' Overall caption was difficult to interpret

Thank you for your suggestion, the caption has been revised (lines 356-371).

Supplementary material:

discussion of detrital analyses is minimal

We have included the analytical procedure for the detrital samples at lines 379-382.

citation call outs are confusing

The footnotes have been replaced by reference numbers.

there seems to be a difference in statement criteria for including a site (5 pts. in text, 4 pts. in SM).

Thank you, this has been changed to four at line 339.

not clear data in SF1 support claims made in text, lines and errors envelopes (?) not described, Cape Verde typo in legend, are all data from leached samples?

Figure S1 has been moved to the main manuscript as Figure 1 and the caption has been revised to include the shaded areas.

table footnotes are not clear... e.g., 'of "TD" indicate...?', 'samples were combined'?

The footnotes have been revised (lines 459-466).

Reviewer #2 (Remarks to the Author):

This is the review of the paper written by Batenburg et al. and submitted to Nature Communications. I must first apologize for having retained the manuscript because of my late review. I also have to clearly state that my main domain of expertise is on climate modeling. Likewise, my review should be seen as one from a non-specialist of Nd. The paper focuses on Nd isotope records over the 72-56 Ma period from Rio Grande Rise, Walvis Ridge (South Atlantic) and Newfoundland basin (North). Measurements have been made on Fe-Mn coatings of sediments and on detrital fractions when it was possible. This technic should allow in principle to disentangle the oceanographic from the continental source signal as Nd isotopes record both signal. Whatever the interpretations the authors have of the similar trend followed by both coatings and detrital fraction, the 4 IODP coatings records in Figure 2 show a convergence toward values comprised between -10 and -8. The convergence trend starts at 65-63 Ma and is complete at 59-58 Ma. The authors use these synchronous trends to propose the onset of more intense deep-water circulation in the Atlantic basin, which in turn, should explain the fact that Nd isotope values become closer whatever their geographical distribution were in the Atlantic Ocean. Although I do agree with these conclusions, the authors go a step further and discuss the potential implications of a well-mixed Atlantic ocean. At this time, the paper becomes more hypothetical. Based on a study of Thomas et al. (2014), the authors suggest that a more ventilated deep ocean (less restricted) may increase the ocean heat transport which in turn would feedback on the thermal latitudinal gradient and on the global climate (and temperature). They then suggest that the early warming phase of the Paleogene may be ascribed to the intensification of deep-water circulation.

In the paper of Thomas et al. (2014), using early Eocene boundary conditions, the early Cenozoic circulation simulated by the MITgcm is rather sluggish and it is only by artificially increasing the vertical mixing coefficient by 50% (potentially attributable to changes in tidal mixing) and then by 500 % at every depth (very unlikely or at least unconstrained) that Thomas et al. are able to simulate an intense oceanic circulation in the Pacific Ocean. So, in principle, it is not the changing paleogeography that induces better-ventilated water in the Thomas et al.'s paper but rather some other processes. For a small part, tidal mixing, for a larger part, severe storms and tropical cyclones hypothesized to occurred more frequently in a warmer world may have provided the energy required by the ocean for a global increase in its mixing capability not the evolution of the paleogeography. In that sense, it is the global warming occurring during the early Paleogene that may have induced larger mixing rate in the ocean at that time.

Thank you for your insights into the modelling results by Thomas et al (2014), and into the underlying climatic processes that may have led to enhanced oceanic mixing independently of paleogeography. We have included the following text in lines 182 to 186 on the potential effect of an increase in vertical mixing on ocean circulation: "Global warming may itself have enhanced vertical mixing through increased occurrence of storms and cyclones (Thomas et al., 2014) that enabled more efficient overturning circulation in the Atlantic Ocean. Both the deepening of the Rio Grande Rise and enhanced mixing associated with global warming would have increased the capacity of the overturning circulation in the Atlantic Ocean to transport heat."

While the authors do focus on the convergence trend in their Nd records, the common increasing and decreasing trends have not been under scrutiny (Fig. 2). A decrease from 70 to 65 Ma then an increase from 65 to 59 Ma and then again a decrease from 59 to 50 Ma are clearly visible. I don't know if it is meaningful but this may be of interest if explanations exist.

The directions of the main trends in the data and their possible causes are now described in more detail, with text added at lines 125-145. We observe a trend of decreasing $\epsilon_{Nd(t)}$ from approximately 70 to 63 Ma, an increasing trend from approximately 64 to 60 Ma, and a decreasing trend from 59 Ma onwards. We interpret these changes with the deepening of the sites, changes in radiogenic and unradiogenic sources of Nd, and the increased volumetric exchange of deep water.

In general, I find the paper very well written but too short in terms of explanations or at least uneasy to follow because many hypotheses are done. As there is room for figures and text in Nature Communications, I suggest to the authors to bring back the 2 supplementary figures in the main text and in general to extend discussions whenever it is possible. This paper represents a strong analytical work with many measurements but it fails to be fully informative because many results have been left aside. A good synthesis of what has been done till now is lacking. In the Figure 1, you show sites on which Nd data are existing and sites for which new data have been acquired (U1403, 369, 1267, 525). The link between this figure and the next one is not clear at all. On Figure 2, you are now referring to Rio Grande (516F), Walvis Ridge (two sites, 525 and 1267) and Newfoundland basin (1403). So, site 369 has disappeared and is now site 516F, meaning that you have 3 South Atlantic sites very close to each other and only one in the North Atlantic. This makes all your interpretations in term of deep-water masses less convincing. In addition, why all data already existing are not plotted on Figure 2 with your new data? It is not clear to me. Because, you are showing all pre-existing data on Figure 1 (all orange circles), it is important to plot their Nd values against your own data? This could make your argument for a homogenous deep-water masse expanding in the whole Atlantic Ocean more convincing.

Thank you for your suggestions to improve the presentation of the figures. We have moved the figures from the supplementary information to the main manuscript. The literature data are presented in Figure 1 and discussed in lines 67-87. We have adjusted the map (originally Figure 1, now Figure 2) to be in closer correspondence with the presentation of the new Nd-isotope data now in Figures 3 (originally Figure S2) and 4 (originally Figure 2). Figure 4 provides a synthesis of our findings and now includes the highest resolution data records available for the Paleocene from Demerara Rise and Maud Rise, as well as the evolution of atmospheric CO₂ levels. We hope that these adjustments allow the reader to better assess what has been done before and how our data fill a gap in understanding.

As a consequence, I would like to see eventually this article published in Nature Communications but in the present state, I don't think that it crosses the threshold to be published. My recommendation is to clarify all various points putted forward in this review, not simply in the response to reviewers, but directly within the main text, in order to make this paper more meaningful, accounting for previous works as much as possible and

providing a clear statement of science advances get through new data presented here.

Below are my specific comments:

Abstract:

Ref. 2 and 3 are not appropriate, the latter being on Cenozoic CO₂ reconstruction and not on Late Cretaceous, and the former being a bit at odd with what has been achieved today. I would advice the paper by Brune et al. (2017, Nature Geoscience) that makes a strong point on correlating the decreasing length of continental rifting with atmospheric CO₂ level during the Late Cretaceous. Another study (Puc at et al., 2004, Geology) showed that the Late Cretaceous cooling was occurring uniformly at the latitudes 10-50° strongly suggesting a general decrease of the atmospheric CO₂ level as an explanation rather than a reorganization of the ocean-atmosphere dynamics.

We have included references to the Brune et al. (2017) paper in line 40 and to the Puc at et al., (2003) paper in line 35.

Lines 38. You should consider adding the Nature paper by Gutjahr, A., Ridgwell an co-authors in Nature, 2017

Thank you, we have added this reference.

Lines 66. Ref should be 11-17?

We have adjusted the referencing.

Lines 64-66. The figure S1 without detailed explanations is hard to read. Please provide more details in the text and not simply in the legend. You also give 6 references here but it is hard to see what we need to keep as a message? > It is complicated and there are many hypotheses.

Figure S1 has been moved to the main manuscript and the following text has been added at lines 93-97 of the manuscript to indicate the main message: "A compilation of existing $\epsilon_{Nd(t)}$ records for the period of time from 72 to 50 Ma (Fig. 1) shows that Nd-isotope data for the Paleocene are only available from a limited number of sites and, with the exception of Demerara Rise (MacLeod et al., 2011; Martin et al., 2012), are of limited resolution (less than one sample per two million years). The figure caption (lines 338-343) has been adjusted accordingly.

In addition, in the following sentences, you suggest that Nd signatures fall within a narrow range of -10 to -8 in the Atlantic basin. Your study thus confirms this narrow range but allows going more away back in time? Am I right? It is something I had trouble to decide. Is your study providing something new in terms of understanding of the evolution of the global circulation? Or is your study confirming other results? It is probably because you have condensate the manuscript a lot that I am not really able to deal with this question.

Our new data enable us to follow the evolution of ocean circulation in the Paleogene and to determine when values converged. The added text at lines 93-100 of the manuscript points out that there was a lack of data and a gap in understanding. Our new data fill this gap and provide new insights into the evolution of North Atlantic circulation.

The converging trends in Nd-isotope data in the Paleocene have not been observed before. This is also true for the close correspondence in timing between the establishment of the narrow range in Nd-isotope signatures and the turning point in the temperature evolution of the deep ocean, both at 59 Ma.

Lines 71-73. Figure S1 show more Nd data, I don't see the point made here concerning the enlarging of the Atlantic Ocean and the modern THC-like? Is there a link between Fig. S1 and all references cited just before?

This was a mistake from our side that was also picked up by Reviewer 1. We have replaced it by a reference to Donnadiou et al. (2016).

Lines 84-94. This is where you state that Nd data do record an oceanographic signal and not a continental-weathering signal. I am not sure to get your point here? How is it possible to conclude that your record reproduces the oceanographic signal if both detrital and coatings follow the same trend? From Figure S2, several long-term trends in Nd are visible on both records although significantly offset as you do recognize. In addition, your argument that offshore records far enough from the coast cannot be influenced by detrital inputs is plausible but then why the detrital record follows the coating's one?

After a discussion with K. Tachikawa and E. Puecat, I think you should consider extending the discussion here to suggest that authigenic sediment may explain why detrital records follow the coating's one (see Moiroud et al., 2016 and Tachikawa et al., 2016)

We have rewritten the section on the detrital results and have included a more detailed discussion of the parallel trends between seawater and detrital Nd-isotope values in lines 109-117. We have followed the reviewer's suggestion and have included the suggested mechanism through which authigenic clays may incorporate dissolved Nd in l.115-117: "In addition, the dissolved seawater Nd-isotope signature may have been incorporated into the hydrogenous component of pelagic clays (Goldstein and O'Nions, 1981; Moiroud et al., 2015), which may partly explain the similarity in the long-term evolution of the detrital and leached $\epsilon\text{Nd}(t)$ values."

Lines 124-127. Do you see an increased difference between detrital and coatings values across the time period you are working on? If not, this sentence is not supported by your own data.

We have now included text at lines 111-115 to explain that although our data faithfully record seawater signatures, the Nd-isotope composition of the water-masses themselves may have been influenced by local factors such as boundary exchange processes. At line 112 we have added "The Nd-isotope composition of the water-masses themselves may have been influenced to some extent by local factors" to indicate that we are not referring to the difference between detrital fractions and ferromanganese coatings, but rather the influence of detrital inputs on the seawater inventory of Nd.

Lines 132-134. I don't agree, the convergence starts earlier while temperature were cooling from 65 to 59 Ma

We agree with this observation also made by Reviewer 1. We realized that we had not made a distinction between the process of convergence and the moment when values had converged. We have replaced "Nd isotope convergence" by "the close correspondence in Nd isotope values" in lines 171 and 154.

Lines 142. "Together with increasing atmospheric CO₂ levels³, the changing paleogeography of the Atlantic thus likely contributed significantly to the boundary conditions that pushed the Earth's climate into a greenhouse state."

The paper, ref.3 you are referring to, suggest that atmospheric CO₂ levels decreased while the climate was cooling from 54 to 38 Ma. I am sure there are other papers that CO₂-proxies showed an increase at this time. This reference is not the right one.

This is correct and we have replaced this reference by "(Brune et al., 2017; Foster et al., 2017)" at line 176.

Figure 2: if Rio Grande Rise is so close to Walvis Ridge, why so much differences in Nd Maastrichtian values? What is the history of depth of these sites within the time interval ? Is there a possibility that you are also recording the deepening of your IODP sites ?

Rio Grande Rise is not very close to the Walvis Ridge as is now indicated on the revised map. Sites 525 and 516, however, are at a similarly shallow water depth, and Site 525 is very close to deeper site 1267. The differences between sites are now addressed explicitly in lines 128-133: "Nd-isotope values at Site 525 were positively offset from $\epsilon_{Nd(t)}$ signatures at comparably shallow Site 516 on the Rio Grande Rise and nearby deeper Site 1267 at the base of the north-western slope of the Walvis Ridge until the end of the Cretaceous. This positive offset was most likely caused by the weathering influx of volcanic material from the partially subaerially exposed Walvis Ridge in the latest Cretaceous (Muller et al., 2008; Pérez-Díaz and Eagles, 2017). The offset decreased as the ridge and Site 525 subsided."

The effect of deepening has been clarified in the discussion of long term increasing and decreasing trends in the latest Cretaceous and early Paleocene in lines 134-140.

Reviewer #1 (Remarks to the Author):

The authors have undertaken a thorough and thoughtful review. The revised manuscript reads well and adequately addresses the points I raised in my initial review as well as those from the other review. I suggest a few minor corrections below but otherwise recommend publication.

22, 57- delete 'radiogenic'

47- add Frank and Arthur, 1999, Paleoceanography to references

126- replace 'most negative' with 'lower'

135- assign this trend [add word]

171- The close correspondence in Nd-isotope value among sites at 59 Ma [add words]

Reviewer #2 (Remarks to the Author):

Dear authors,

I have read with a great interest the revision of your paper. You have extensively rewritten your paper accounting for comments made on the first round. Except for one mistake in the referencing - ref 9 should be Puceat et al., 2007, Geology and not the 2003 paper in Paleoceanography - I have no reasons to retain this paper and I can recommend the publication.

Response letter

Dear editorial team,

We are grateful for the careful evaluation of our resubmitted manuscript by the reviewers. The following is a detailed response (in blue) to the reviewers' comments (in black)

Reviewer #1 (Remarks to the Author):

The authors have undertaken a thorough and thoughtful review. The revised manuscript reads well and adequately addresses the points I raised in my initial review as well as those from the other review. I suggest a few minor corrections below but otherwise recommend publication.

22, 57- delete 'radiogenic'

Three occurrences of the word "radiogenic" have been deleted (l. 21, l. 50, l. 55).

47- add Frank and Arthur, 1999, Paleoceanography to references

This reference has been added to the references (l. 45, and References section).

126- replace 'most negative' with 'lower'

We have replaced "most negative" with "lowest" (l. 126)

135- assign this trend [add word]

The word "trend" has been added (l. 135).

171- The close correspondence in Nd-isotope value among sites at 59 Ma [add words]

The words "among sites" have been added (l. 170).

Reviewer #2 (Remarks to the Author):

Dear authors,

I have read with a great interest the revision of your paper. You have extensively rewritten your paper accounting for comments made on the first round. Except for one mistake in the referencing -ref 9 should be Puceat et al., 2007, Geology and not the 2003 paper in Paleoceanography - I have no reasons to retain this paper and I can recommend the publication.

Thank you, the incorrect reference has been replaced by the 2007 reference (l. 270).

We hope that the changes address all issues raised. Thank you for considering our manuscript for publication.

Kind regards, on behalf of all authors,

Sietske Batenburg